# Fetal and Perinatal Outcome Following First and Second Trimester COVID-19 Infection: Evidence from a Prospective Cohort Study

**DOI:** 10.3390/jcm10102152

**Published:** 2021-05-16

**Authors:** Hadar Rosen, Yossi Bart, Rita Zlatkin, Liat Ben-Sira, Dafna Ben Bashat, Sharon Amit, Carmit Cohen, Gili Regev-Yochay, Yoav Yinon

**Affiliations:** 1Fetal Medicine Unit, Department of Obstetrics and Gynecology, Sheba Medical Center, Tel-Hashomer 5262000, Israel; rosenhadar@gmail.com (H.R.); yossib86@gmail.com (Y.B.); ritaz123@gmail.com (R.Z.); 2Division of Pediatric Radiology, Tel Aviv Sourasky Medical Center, Tel Aviv 6423906, Israel; bensiraliat@gmail.com; 3Sackler School of Medicine, Tel-Aviv University, Tel-Aviv 69978, Israel; dafnab@tlvmc.gov.il (D.B.B.); Sharon.Amit@sheba.health.gov.il (S.A.); Carmit.Cohen@sheba.health.gov.il (C.C.); Gili.Regev@sheba.health.gov.il (G.R.-Y.); 4Sagol School of Neuroscience, Tel Aviv University, Tel-Aviv 69978, Israel; 5Sagol Brain Institute, Tel Aviv Sourasky Medical Center, Tel Aviv 64239, Israel; 6Clinical Microbiology, Sheba Medical Center, Tel-Hashomer 5262000, Israel; 7Infection Prevention & Control Unit, Sheba Medical Center, Tel-Hashomer 5262000, Israel

**Keywords:** first and second trimester, maternal infection, COVID-19, vertical transmission, fetal outcome

## Abstract

A novel coronavirus termed severe acute respiratory syndrome coronavirus 2 (SARS-CoV-2) is a new strain of coronavirus causing coronavirus disease 2019 (COVID-19) disease, which emerged as a global pandemic. Data regarding the implications of COVID-19 disease at early gestation on fetal and obstetric outcomes is scarce. Thus, our aim was to investigate the effect of first and second trimester maternal COVID-19 disease on fetal and perinatal outcomes. This was a prospective cohort study of pregnant women with a laboratory-proven SARS-COV-2 infection contracted prior to 26 weeks gestation. Women were followed at a single tertiary medical center by serial sonographic examinations every 4–6 weeks to assess fetal well-being, growth, placental function, anatomic evaluation and signs of fetal infection. Amniocentesis was offered to assess amniotic fluid SARS-COV-2-PCR (polymerase chain reaction) and fetal brain magnetic resonance imaging (MRI) was offered at 30–32 weeks gestation. Demographic, obstetric and neonatal data were collected from history intake, medical charts or by telephone survey. Perinatal outcomes were compared between women infected at first vs. second trimester. 55 women with documented COVID-19 disease at early gestation were included and followed at our center. The mean maternal age was 29.6 ± 6.2 years and the mean gestational age at viral infection was 14.2 ± 6.7 weeks with 28 (51%) women infected at the first trimester and 27 (49%) at the second trimester. All patients but one experienced asymptomatic to mild symptoms. Of 22 patients who underwent amniocentesis, none had evidence of vertical transmission. None of the fetuses exhibited signs of central nervous system (CNS) disease, growth restriction and placental dysfunction on serial ultrasound examinations and fetal MRI. Pregnancies resulted in perinatal survival of 100% to date with mean gestational age at delivery of 38.6 ± 3.0 weeks and preterm birth <37 weeks rate of 3.4%. The mean birthweight was 3260 ± 411 g with no cases of small for gestational age infants. The obstetric and neonatal outcomes were similar among first vs. second trimester infection groups. We conclude SARS-CoV-2 infection at early gestation was not associated with vertical transmission and resulted in favorable obstetric and neonatal outcomes.

## 1. Introduction

Over the past year, the coronavirus disease 2019 (COVID-19) pandemic outbreak has emerged with global numbers constantly on the rise [1] The novel severe acute respiratory syndrome coronavirus 2 (SARS-CoV-2) may affect pregnant women at any stage in pregnancy, and can potentially have negative effects on both mother and child.

Perinatal outcomes resulting from maternal viral infections during pregnancy range anywhere from no effect to pregnancy loss to fetal infection with resulting congenital viral syndromes, as well as various obstetrical complications [2,3]. Recent examples of new viral infections that showed substantial pregnancy implications are the 2009 pandemic H1N1 influenza virus [2] and the severe fetal effects of Zika virus [3]. Accumulating evidence suggests that pregnant women with COVID-19 are at increased risk for severe maternal outcomes compared to non-pregnant women including increased risk for intensive-care unit (ICU) admission, invasive ventilation and death [4].

Moreover, COVID-19 in pregnancy was found to be associated with increased risk for preterm birth, although mostly iatrogenic and not spontaneous preterm birth [5,6].

However, most of these data were based on cases of COVID-19 which occurred in the late second or third trimester of pregnancy. 

From the fetal perspective, another interesting issue involves the risk of vertical transmission to the fetus, if any. To date, publications reporting vertical transmission rates of SARS-CoV-2 refer mostly to third trimester infections rather than early pregnancy ones [7,8,9]. Possible cases of congenital infection have been reported in the setting of third-trimester maternal infection suggesting that congenital infection is uncommon occurring in approximately 2% of maternal infections [10]. However, vertical transmission following first or second trimester infection have not been reported to date. 

While it is known that some maternal viral infections such as Cytomegalovirus contracted at late gestation usually do not lead to fetal sequelae, acquiring infection at early gestation may inflict harm on the fetus. Nevertheless, limited information if any exists in the literature as to the fetal implications of acquiring SARS-CoV-2 infection during the first and second trimester of pregnancy. One study reported on no significant increase in the early pregnancy loss rate [11]. Thus, the effect of SARS CoV-2 infection at early gestation on the remainder of the pregnancy and on fetal wellbeing is unknown. 

With this lack of data in mind, we sought to comprehensively follow pregnant women with a laboratory proven first or second trimester SARS-COV-2 infection and investigate vertical transmission rates, fetal adverse effects, obstetrics complications and delivery outcomes.

## 2. Methods

### 2.1. Study Population

This was a prospective cohort study including pregnant women with laboratory confirmed SARS-CoV-2 infection prior to 26 weeks of gestation. The diagnosis of SARS-CoV-2 infection was based on a positive result on real-time reverse-transcriptase polymerase-chain-reaction (RT-PCR) assay of nasal and pharyngeal swab specimens. Women were tested due to symptoms or exposure. During the study period, we conducted maternal serum SARS CoV-2 IgG serology screening studies for women undergoing routine amniocentesis at around 17–20 weeks gestation in an attempt to identify women who experienced asymptomatic SARS-COV-2 infection during pregnancy. Out of 155 gravid women tested, we found 3 who tested positive and were also included in our cohort. Following their recovery from the acute viral illness phase, women were referred to and followed at the Fetal Medicine Unit at Sheba Medical Center between March 2020 and February 2021. The cohort was divided into two groups according to the timing of infection: patients who acquired the infection prior to 13 + 6 weeks of gestation were defined as first trimester COVID-19 disease, and those who acquired the infection between 14–26 weeks of gestation were defined as second trimester COVID-19 disease. After signing informed consent, patients were followed at our clinic throughout the rest of the pregnancy as well as during the immediate post-partum period. Delivery data for women who delivered in other medical centers was completed by telephone survey and a review of medical records.

The study was approved by the Sheba Medical Center Institutional Review Board (REB 7122-20-SMC).

### 2.2. Data Collection

Data were collected prospectively at the first visit and included the following demographic and obstetrical characteristics: maternal age, gravity, parity, previous medical, surgical and obstetric history. Pregnancy information including verification of gestational age by initial first trimester ultrasound as well as prenatal screening tests and fetal anatomy scans completed along pregnancy were reviewed and recorded. We then collected data regarding the acute viral infection such as gestational age upon attaining the virus, acute COVID-19 illness phase severity, symptoms, need for hospitalization and medications administered. 

### 2.3. Follow up of SARS-CoV-2 Infected Pregnant Women 

Following initial recruitment, our study focused on three main areas of investigation:Vertical transmission rates—all women in our cohort were offered amniocentesis for amniotic fluid RT-PCR test of SARS CoV-2 at the time of genetic amniocentesis. Those who opted for the test, were tested for amniotic fluid SARS CoV-2 RT-PCR as well as maternal serum serology for SARS CoV-2 IgG and IgM.Fetal imaging throughout pregnancy—women were followed with serial ultrasound investigations looking for sonographic evidence of fetal disease secondary to viral infection from the time of recruitment and every 4–6 weeks thereafter. To complete fetal brain evaluation, we offered a fetal brain magnetic resonance (MR) study at 30–32 weeks gestation.Delivery and newborn data were collected from medical charts or by telephone survey. Small for gestational age (SGA) was defined as birthweight <10th percentile. When possible, we collected umbilical cord blood and tested for SARS CoV-2 IgG and IgM antibodies.

The follow up timeline is depicted in Figure 1. 

### 2.4. Serology

Maternal and Fetal cord blood were centrifuge at 4000 G for five minutes at room temperature and sera ware used for a semi-quantitative enzyme-linked immunosorbent assay (ELISA). IgG and IgM expression against RBD spike antigen was tested as previously reported [12]. ELISA index value was defined as the ratio between sample and cut-off Optical Densities. ELISA index value above 1.1 were considered positive for analysis.

### 2.5. Statistical Analysis

The normality of the data was tested using the Shapiro-Wilk or Kolmogorov-Smirnov tests. Data are presented as median and inter-quartile range (IQR) or mean and standard deviation (SD). Comparison between continuous variables was conducted with Student’s *t*-test or Mann–Whitney U test, as appropriate. The chi-square and Fisher’s exact tests were used for comparison between categorical variables. Significance was accepted at *p* < 0.05. Statistical analyses were conducted using the IBM Statistical Package for the Social Sciences (IBM SPSS v.27; IBM Corporation Inc., Armonk, NY, USA).

## 3. Results 

From March 2020 to February 2021 55 women were followed at the Sheba Fetal Medicine Unit following early pregnancy laboratory confirmed SARS-CoV-2 infection. The cohort demographic and clinical data are shown in Table 1. 

Fifty-one were singleton pregnancies and 4 were twin pregnancies. The mean maternal age was 29.6 ± 6.2 years and the mean gestational age at viral infection was 14.2 ± 6.7 weeks of gestation with 28 (51%) women infected at the first trimester and 27 (49%) at the second trimester. All patients but one experienced asymptomatic to mild symptoms as classified by the NIH [13]. Six patients required hospitalization for observation due to shortness of breath, yet none of the patients required assisted ventilation. Hospitalized patients received supportive therapy only and did not require antiviral therapy. The specific symptoms experienced by the patients are detailed in Table 1. One patient was hospitalized with severe myocarditis 1 week after illness recovery, which has resolved. 

When comparing between first and second trimester infection groups, most demographic and clinical variables were similar among the two groups except pre-pregnancy Body Mass Index (BMI) which was lower in the first trimester patients (22.1 ± 2.1 vs. 24.9 ± 5.1 respectively *p* < 0.01). Moreover, first trimester patients complained more often on headaches (*p* = 0.02), whereas shortness of breath was experienced more commonly by second trimester patients although not statistically significant (Table 1). 

### 3.1. Vertical Transmission Rates

The fetal investigation studies are detailed in Table 2. In 22 women whom amniocentesis was performed for fetal genetic microarray testing, amniotic fluid SARS CoV-2 RT-PCR was also tested. No genetic aberrations were detected in our cohort. All 22 amniotic fluid SARS CoV-2 RT-PCR tests were negative, while 12 (54%) showed evidence of maternal serum IgG.

We obtained 3 samples of fetal blood, of which 2 were sampled due to fetal anomalies (their cases are detailed below) and one was diagnosed with SARS CoV-2 infection upon admission for an intra-uterine transfusion due to RH isoimmunization. All fetal blood SARS CoV-2 RT-PCR as well as IgM for SARS CoV-2 results were negative.

Fetal cord blood sampling at delivery was available in 4 of the cases. IgG antibodies for SARS CoV-2 were found in 3 of the cases but IgM antibodies were not detected, suggesting placental transfer of maternal antibodies rather than vertical transmission with fetal infection.

### 3.2. Fetal Imaging Findings Along Pregnancy

Of the participating women, 38 were followed at our center with serial targeted sonographic scans every 4–6 weeks during pregnancy. There were 2 cases of major anomalies detected that will be outlined.

In one case where the mother was infected with SARS CoV-2 at 8 weeks of gestation, major fetal anomalies were detected including anhydramnios and small echogenic kidneys. In an attempt to understand whether these findings were related to fetal infection, we completed a thorough investigation including fetal blood sampling which revealed no genetic abnormalities on chromosomal microarray analysis (CMA) testing, and negative SARS CoV-2 RT-PCR. Other possible maternal viral infections were also ruled out. Due to severe early anhydramnios and dismal renal prognosis the patient opted for pregnancy termination, during which a fetal blood sample was sent and found negative for SARS CoV-2 IgM antibodies; hence we could not verify fetal SARS CoV-2 infection as a possible cause for the fetal anomalies.

Another case of a 31-year-old G2P1 who was diagnosed with mild symptomatic COVID-19 disease at 15 weeks of gestation, an isolated finding of unilateral fetal cataract was detected at a routine anatomy scan at 23 + 3 weeks of gestation. Routine viral workup was negative. The couple opted for pregnancy termination and a fetal blood sample was obtained for SARS CoV-2 IgM which was negative. 

Interestingly, we were able to complete 6 fetal brain MRI studies at 32–36 weeks gestation, indicating normal brain development without evidence of abnormalities and normal range biometric measurements. 

### 3.3. Obstetric Outcome and Neonatal Data

Obstetric outcomes are presented in Table 3. 

At the time of submission, 29 of the 55 women delivered and their data were available for analysis. Mean gestational age at delivery was 38.6 ± 3.0 weeks gestation. There was only 1 (3.4%) preterm birth at 36 weeks gestation. 26 (89.7%) of the women delivered vaginally with 2 (6.9%) operative deliveries. 3 (10.3%) women delivered via cesarean section for common obstetrical reasons unrelated to SARS CoV-2. The mean birthweight was 3260 ± 411 g with no cases of small for gestational age (SGA). All neonates did well on initial neonatologist assessments with normal APGAR scores and cord blood pH. On post-partum follow up we had 1 (3.4%) case of medically treated post partum hemorrhage, 3 (10.3%) cases of postpartum fever and 1 (3.4%) case of arm superficial venous thrombosis at the Intra-venousline site. No differences were noted when assessing labor and delivery outcome measures between first and second trimester infections as shown in Table 3. 

## 4. Discussion

### 4.1. Main Findings 

Our study focused on the implications of maternal SARS CoV-2 infection during the first and second trimesters of pregnancy with regards to vertical transmission rates, fetal effects and obstetric outcomes.

The main findings of our study were:No evidence of vertical SARS-COV-2 transmission was found upon amniotic fluid PCR testing or cord blood serology at delivery.No evidence of adverse fetal effects was found on serial ultrasound anatomy scans, fetal growth as well as a few cases of fetal brain MR imaging.No increased rates of obstetric complications were found. Deliveries occurred at term and newborns were appropriate for gestational age. Neonatal outcome was overall reassuring.There was no significant difference for the above outcome measures when comparing first to second trimester infections.

To date, information regarding the impact of maternal SARS CoV-2 infection at early gestation on fetal and pregnancy outcome is limited. In contrast to our reassuring findings regarding obstetric and perinatal outcomes, a multicenter study including 388 pregnant women from 73 centers with confirmed SARS CoV-2 infection reported on a 26.3% rate of preterm birth [14]. However, 80% of preterm deliveries were indicated resulting in only 5.2% rate of spontaneous preterm birth. Of note, most cases included in that study were diagnosed with SARS CoV-2 in the third trimester. In accordance with our findings, the risk of vertical transmission in their study was negligible, with only one case confirmed positive after delivery. Despite evidence indicating a low rate of vertical transmission of 1–2% [15], one major unresolved concern regarding SARS CoV-2 in pregnancy is whether it is associated with adverse fetal outcome including miscarriage, intra-uterine fetal death and placental complications. The rate of stillbirths and neonatal deaths has been reported to date to be slightly increased, although most neonatal deaths were related to prematurity [16]. In the aforementioned multicenter study there were 6 miscarriages (2.3%), six intrauterine deaths (2.3%) and 5 (2%) neonatal deaths with an overall perinatal death rate of 4.2%. A secondary analysis of this study revealed that the incidence of composite adverse fetal outcome was significantly higher when the infection occurred in the first trimester [17]. Our study does not support this observation as we found no adverse fetal outcome following first and second trimester SARS CoV-2 infection. Moreover, through a very close fetal surveillance protocol including ultrasound evaluation and Doppler studies every 4 weeks, we did not demonstrate any evidence for placental dysfunction and placenta-related pathologies such as intra-uterine growth restriction and preeclampsia. These data are not in line with previous reports indicating that COVID-19 in term patients was associated with increased rates of placental histopathological abnormalities including fetal vascular malperfusion and villitis of unknown etiology. However, based on our study we assume that these reported placental findings in COVID-19 patients are not translated into a clinical placenta-mediated disease in the majority of cases. Furthermore, due to the raised concern with respect to the fetal outcome including potential neurological manifestations following transplacental transmission of SARS-CoV-2 infection [18], we evaluated our patients throughout pregnancy with repeated neurosonography examinations including fetal brain MRI in 6 cases without evidence of any fetal CNS abnormality. 

### 4.2. Strengths and Limitations

The strength of our study stems from the novelty of the data presented and its prospective design. To our knowledge, this is the first cohort of early gestational COVID-19 affected pregnancies that underwent prospectively longitudinal and comprehensive evaluation along pregnancy including amniocentesis to detect vertical transmission and repeated thorough ultrasound examinations including Doppler studies and neurosonography. 

A few obstacles may hinder our results. 

Our cohort is preliminary in size and results need to be repeated in larger cohort studies as the pandemic continues to affect more early pregnancies worldwide. Moreover, since the incidence of adverse fetal outcome is low, our sample size is probably underpowered to draw definite conclusions regarding the true risk of miscarriage or fetal death following maternal COVID-19 disease in early pregnancy. Of note, given the low number of women with moderate or severe COVID-19 disease in our cohort, we cannot estimate the impact of severe disease at early gestation on pregnancy outcome. 

We found the methodology proving vertical transmission during pregnancy challenging. All amniotic fluid samples, which were obtained for SARS CoV-2 RT-PCR resulted negative, yet urine may not be the optimal medium to test for detection of fetal infection with SARS CoV-2. In neonates, according to CDC guidelines, a nasopharynx, oropharynx, or nasal swab sample are used to determine infection rather than a urine sample [19], but obtaining such samples is not practical from fetuses in utero. We analyzed 3 fetal blood samples for SARS CoV-2 RT-PCR and IgM serology, and those were negative as well. 

This report is an attempt to bridge a gap in literature and provide important information for ongoing as well as future planned pregnancies during this pandemic. Our data may serve to reassure expecting parents as well as providers that COVID-19 infections in early pregnancy do not pose additional fetal, obstetrical or neonatal risk. 

Future larger-scale studies are due to support the preliminary data presented. 

## 5. Conclusions

Our study did not demonstrate vertical transmission following early pregnancy SARS CoV-2 infection, nor did we encounter fetal, obstetric or neonatal adverse outcomes that could be attributed to COVID-19 disease at early gestation. This is a reassuring preliminary study yet large-scale early pregnancy COVID-19 disease cohorts are needed to affirm our findings.

## Figures and Tables

**Figure 1 jcm-10-02152-f001:**
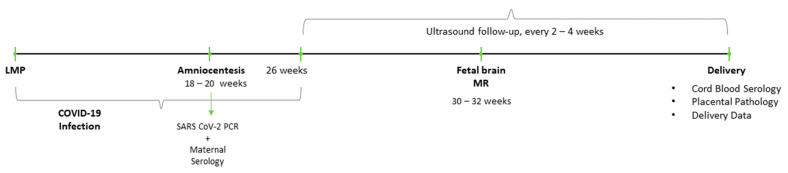
Planned follow up timeline for a cohort of pregnancies contracting early (prior to 26 weeks gestation) coronavirus disease 2019 (COVID-19) infection.

**Table 1 jcm-10-02152-t001:** Demographics and clinical characteristics of women with first or second trimester COVID-19 infection.

Variable	Overall (*n* = 55)	First Trimester Infection (*n* = 28)	Second Trimester Infection (*n* = 27)	*p*-Value
Age (year mean ± SD)	29.6 ± 6.2	28.9 ± 6.0	30.3 ± 6.4	0.30
Gravity	2 (1–5)	2 (1.2–4)	2.5 (1–5)	0.52
Parity	1 (0–3)	1 (0–2.7)	1 (0–3.2)	0.70
Singleton	51 (92.7)	24 (85.7)	27 (100)	0.04
BMI	23.6 ± 4.1	22.1 ± 2.1	24.9 ± 5.1	<0.01
19–25	23 (71.9)	15 (93.8)	8 (50.0)	<0.01
25–30	6 (18.8)	1 (6.3)	5 (31.3)	0.07
Gestational week at infection (avg)	14.2 ± 6.7	8.4 ± 3.3	20.1 ± 3.2	<0.01
Severity of symptoms				
Asymptomatic	5 (9.6)	1 (3.8)	4 (15.4)	0.16
Mild	42 (80.8)	23 (88.5)	19 (73.1)	0.16
Moderate	4 (7.7)	2 (7.7)	2 (7.7)	1.00
Severe	1 (1.9)	0	1 (3.8)	0.31
Length of illness (days)	17.8 ± 13.5	15.6 ± 11.7	20.4 ± 15.2	0.28
Symptoms				
Fever	16 (31.4)	7 (28.0)	9 (34.6)	0.61
Shortness of breath	6 (11.5)	1 (3.8)	5 (19.2)	0.08
Cough	14 (26.9)	5 (19.2)	9 (34.6)	0.21
Fatigue	27 (51.9)	15 (57.7)	12 (46.2)	0.41
Anorexia	3 (6.3)	2 (8.3)	1 (4.2)	0.55
Loss of smell/taste	29 (55.8)	17 (65.4)	12 (46.2)	0.16
Myalgia	14 (26.9)	6 (23.1)	8 (30.8)	0.53
Headache	15 (29.4)	11 (44.0)	4 (15.4)	0.02
Hospitalization	6 (11.1)	4 (14.8)	2 (7.4)	0.39

Data are shown as median (interquartile range (IQR)), mean (± standard deviation (SD)) or number (percentage), BMI, Body Mass Index.

**Table 2 jcm-10-02152-t002:** Fetal investigations during pregnancy.

Investigation	Number of Patients	Results	
		Positive Findings	Negative Findings
Amniocentesis			
Amniotic fluid COVID-19 PCR	22	0	22
Fetal Brain MRI	5	0	5
Fetal anomaly scans	38	8 Ɨ	30
Cord blood at delivery			
IgM	4	0	
IgG	4	3	

Ɨ see detailed in text.

**Table 3 jcm-10-02152-t003:** Obstetric and perinatal outcomes of women with first or second trimester COVID-19 infection.

	Overall	First Trimester COVID (*n* = 7)	Second Trimester COVID (*n* = 22)	*p*-Value
Perinatal survival	29 (100)	7 (100)	22 (100)	1.00
Gestational week at delivery	38.6 ± 3.0	39.3 ± 1.6	38.3 ± 3.3	0.67
Preterm birth	1 (3.4)	0 (0)	1 (4.3)	
Mode of delivery				
Vaginal	24 (82.8)	6 (85.7)	18 (81.8)	1.00
Operative	2 (6.9)	1 (14.3)	1 (4.5)	0.43
Cesarean Section	3 (10.3)	0	3 (13.6)	0.56
Labor induction	6 (21.4)	0	6 (28.6)	0.29
Birthweight	3260 ± 411	3132 ± 449	3301 ± 400	0.69
Apgar				
1 min	9 (9–9)	9 (9–9)	9 (9–9)	1.00
5 min	10 (10–10)	10 (10–10)	10 (10–10)	1.00
Arterial pH				
Post-Partum Complication				
PPH	1 (3.4)	0	1 (4.5)	1.00
Fever	3 (10.3)	1 (14.3)	2 (9.1)	1.00
Thromboembolic event	1 (3.6)	0	1 (4.8)	1.00

Data are shown as median (IQR), mean (±SD) or number (percentage). PPH, Post-Partum Hemorrhage.

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
