# Peer review of "Fetal and Perinatal Outcome Following First and Second Trimester COVID-19 Infection: Evidence from a Prospective Cohort Study"

_jcm, 2021, doi:10.3390/jcm10102152_

Round 1

Reviewer 1 Report

This is a well-written manuscript that would add to the existing literature. 

Can the authors comment about the possibility of selection bias given the relatively low number of women with moderate or severe disease? Specifically, could there be an increased risk of miscarriage in those with severe first trimester COVID infection or increased preterm birth in those with severe infections in the late second trimester that were not detected in the cohort?

How many patients required hospitalization during their illness? What were the criteria for hospitalization? What was the recommended therapy, if any, in those undergoing outpatient management?

Thank you so much for the opportunity to review the manuscript titled: “Fetal and perinatal outcome following first and second trimester COVID-19 infection: Evidence from a prospective cohort study” submitted for consideration by Dr. Hadar Rosen and colleagues from Sheba Medical Center/Tel Aviv University, Israel 

The aim of this study was to investigate the effect of first and second trimester maternal COVID-19 disease on fetal and perinatal outcomes. This was a prospective cohort study 
of pregnant women with a laboratory proven COVID-19 disease contracted prior to 26 weeks’ gestation. Women were followed at a single tertiary medical center by serial sonographic examinations every 4-6 weeks to assess fetal well-being, growth, placental function, anatomic evaluation and signs of fetal infection. Amniocentesis was offered to assess amniotic fluid COVID-19 PCR and fetal brain MRI was offered at 30-32 weeks’ gestation. Demographic, 
obstetric and neonatal data were collected from history intake, medical charts or by telephone survey. Perinatal outcomes were compared between women infected at first vs. second trimester. 

55 women with documented COVID-19 disease at early gestation were included and followed at the academic center.  The mean maternal age was 29.6 ±6.2 years and the mean gestational age at viral infection was 14.2 ±6.7 weeks with 28 (51%) women infected at the first 
trimester and 27 (49%) at the second trimester.  All patients but one experienced asymptomatic to mild symptoms. Of 22 patients who underwent amniocentesis, none had evidence of vertical transmission. None of the fetuses exhibited signs of CNS disease, growth restriction and placental dysfunction on serial ultrasound examinations and fetal MRI. Pregnancies resulted in perinatal survival of 100% to date with mean gestational age at delivery of 38.6 ±3.0 weeks and preterm birth < 37 weeks rate of 3.4%.  The mean birthweight was 3260 ±411g with no cases of small for gestational age infants. The obstetric and neonatal outcomes were similar among first vs. second trimester infection groups. The authors therefore concluded that COVID-19 disease at early gestation was not associated with vertical transmission and resulted in 
favorable obstetric and neonatal outcomes.

Overall this is a well-written manuscript that is within the scope of the journal. While the cohort size is relatively small and the study therefore underpowered to detect some of the outcomes, it is appropriate for a single center study with granular outcome data. Moreover, the investigators conducted detailed ultrasound surveillance, MRI imaging, and almost routine amniocentesis to detect vertical transmission. Given the lack of fetal findings the transmission risk is likely small, and therefore these findings are likely generalizable. These data would add to the existing literature and would likely be incorporated into patient counseling. I do have a few minor comments for the investigators and would be delighted to review an updated draft after revision.

Author Response

Can the authors comment about the possibility of selection bias given the relatively low number of women with moderate or severe disease? Specifically, could there be an increased risk of miscarriage in those with severe first trimester COVID infection or increased preterm birth in those with severe infections in the late second trimester that were not detected in the cohort?

  1. At the time the study was conducted (starting March 2020) and patients were recruited severe manifestations for child baring age women were rare. Our recruitment was based on physician referrals and not restricted by the severity of the initial disease; therefore, we believe the indolent symptoms described represent the disease manifestation at that time. Our report is descriptive and depicts the outcome of these pregnancies. While the questions regarding the implications of more severe disease are interesting, we are not able to address them within our cohort.This point has been added to the discussion as follows: “Of note, given the low number of women with moderate or severe COVID-19 disease in our cohort, we cannot estimate the impact of severe disease at early gestation on pregnancy outcome.”

How many patients required hospitalization during their illness? What were the criteria for hospitalization? What was the recommended therapy, if any, in those undergoing outpatient management?

  1. . Six patients required hospitalization bot not required assisted ventilation and did not receive antiviral therapy. This information has been added to the text as follows: “All patients but one experienced asymptomatic to mild symptoms as classified by the NIH. Six patients required hospitalization for observation due to shortness of breath, yet none of the patients required assisted ventilation. Hospitalized patients received supportive therapy only and did not require antiviral therapy. The specific symptoms experienced by the patients are detailed in Table 1. One patient was hospitalized with severe myocarditis 1 week after illness recovery, which has resolved.

Reviewer 2 Report

This work by Rosen et al., reports the fetal and perinatal outcomes following first vs second trimester infection with the SARS-CoV-2 virus in a prospective cohort of 55 women that were followed carefully until the end of their pregnancy. They show that, so far, the impact on pregnancy and fetal outcomes are minimal.

Although this seems in contrast to previously published work reporting important negative impact, these studies are often case reports or very small cohort studies (less then 5 patients) and therefore the selection bias is highly important. Furthermore, the fact that they authors reports strictly on first and second trimester infection is very important considering the current progression of the pandemic worldwide and the number of early pregnancy infection rising. Therefore, I believe this is very important work and, although I have some comments that would improve the work (see below), this definitely warrant publication.  

General comments

  • Throughout the manuscript the term “COVID-19” is used loosely, but COVID-19 refers specifically to the clinical presentation of the disease whilst SARS-CoV-2 is the actual virus. Therefore, there should not be “laboratory proven COVID-19” it should be lab proven SARS-CoV-2. Same for the “COVID-19 PCR” which should be SARS-CoV-2 instead. These are examples from the abstract, but it should be corrected throughout the manuscript.

Specific comments

  • Within the introduction, references should be added to the first sentence of the second paragraph (perinatal outcomes resulting from maternal infections). Fetal loss and obstetrical complications are mentioned without any references even though there are many that could be added.
  • Methods are lacking details:
    • How was maternal serum serology for IgG/IgM done?
    • How was cord blood IgM antibodies detection performed?
  • Figure 1:
    • Change COVID-19 PCR for SARS-CoV-2 PCR.
    • Why does it states “q 2-4 weeks” for the ultrasound follow-up? What does the “q” stands for?
  • Results:
    • Please define abbreviations in full when first used (i.e. BMI, IUT, CMA, etc)
    • In the section on obstetric outcome, it start with “to present, 29 out of 55…” do the authors means that at the time of submission 29 out of the 55 women had delivered? If so, to present should be replaced to clarify the sentence.

Author Response

  1. Throughout the manuscript the term “COVID-19” is used loosely, but COVID-19 refers specifically to the clinical presentation of the disease whilst SARS-CoV-2 is the actual virus. Therefore, there should not be “laboratory proven COVID-19” it should be lab proven SARS-CoV-2. Same for the “COVID-19 PCR” which should be SARS-CoV-2 instead. These are examples from the abstract, but it should be corrected throughout the manuscript.

Reply: These terms have been amended throughout the text accordingly.

  1. Within the introduction, references should be added to the first sentence of the second paragraph (perinatal outcomes resulting from maternal infections). Fetal loss and obstetrical complications are mentioned without any references even though there are many that could be added.

Reply: References have been added to the first sentence accordingly.

  1. How was maternal serum serology for IgG/IgM done?
  2. How was cord blood IgM antibodies detection performed?

Reply to 3+4: At the method section we added a description of the Elisa assay for antibody detection Maternal and Fetal cord blood were centrifuge at 4000 G for five minutes at room temperature and sera ware used for semi quantitative ELISA assay. IgG and IgM expression against RBD spike antigen was tested as previously reported (Indenbaum et al.2020). ELISA index value was defined as the ratio between sample and cut-off ODs. ELISA index value above 1.1 were considered positive for analysis. 

  1. Figure 1 Change COVID-19 PCR for SARS-CoV-2 PCR.

Why does it states “q 2-4 weeks” for the ultrasound follow-up? What does the “q” stands for?

The figure has been corrected as requested

  1. Please define abbreviations in full when first used (i.e. BMI, IUT, CMA, etc)–

Abbreviations were defined throughout the text as requested

  1. In the section on obstetric outcome, it start with “to present, 29 out of 55…” do the authors means that at the time of submission 29 out of the 55 women had delivered? If so, to present should be replaced to clarify the sentence – The term “to present” has been replaced by “At the time of submission” as requested.